# Protein Intake, Protein Mealtime Distribution and Seafood Consumption in Elderly Norwegians: Associations with Physical Function and Strength

**DOI:** 10.3390/geriatrics5040100

**Published:** 2020-12-03

**Authors:** Linda Kornstad Nygård, Lisbeth Dahl, Ingunn Mundal, Jūratė Šaltytė Benth, Anne Marie Mork Rokstad

**Affiliations:** 1Faculty of Health Sciences and Social Care, Molde University College, P.O. Box 2110, 6402 Molde, Norway; ingunn.p.mundal@himolde.no (I.M.); anne.m.m.rokstad@himolde.no (A.M.M.R.); 2Institute of Marine Research (IMR), P.O. Box 1870 Nordnes, 5817 Bergen, Norway; Lisbeth.Dahl@hi.no; 3Institute of Clinical Medicine, Campus Ahus, University of Oslo, P.O. Box 1171, Blindern, 0318 Oslo, Norway; jurate.saltyte-benth@medisin.uio.no; 4Health Services Research Unit, Akershus University Hospital, P.O. Box 1000, 1478 Lørenskog, Norway; 5Norwegian National Advisory Unit on Ageing and Health, Vestfold Hospital Trust, P.O. Box 2136, 3103 Tønsberg, Norway

**Keywords:** nutrition, protein, fish, older adults, physical performance, hand grip strength, sarcopenia, frailty

## Abstract

Protein intake is considered important in the maintenance of muscle health in ageing. However, both the source and mealtime distribution of protein might affect the intake of protein and its effect on muscle protein synthesis. In this study, protein intake, mealtime distribution of protein, and seafood consumption were assessed in 92 older adults (aged 65+), and associations with physical performance (Short Physical Performance Battery (SPPB)), grip strength and gait speed were assessed in a multiple linear regression analysis. The participants had a mean age of 73 ± 8.9 years. Mean protein intake was 1.1 g/kg body weight. Protein intake was well distributed, with coefficient of variance between meals (CV meals) 0.6 ± 0.3. However, dinner had the highest protein intake. No associations were found between the nutrition factors and physical performance or strength; however, this result might have been caused by a ceiling effect in the chosen test batteries, as the mean score on SPPB was 10.3 ± 2.7, and 48.9% of the participants reached the top score of 12 points. Mean grip strength was 44.4 ± 9.4 kg (men) and 26.2 ± 6.8 kg (women). Mean gait speed was 1.0 ± 0.3 m/s. The interaction analysis suggests that there might be gender differences in the effect of seafood consumption on gait speed.

## 1. Introduction

For an older person to preserve good physical function while ageing, the maintenance of muscle mass, muscle function, and muscle strength is of importance. With higher age, more people develop sarcopenia and frailty. Sarcopenia is defined by a loss of muscle mass, strength, and function [1], and is considered a frequent cause of frailty [2]. Frailty is defined as the presence of three or more of the five criteria described by Fried: weight loss, exhaustion, weakness, slowness and inactivity [3]. Recently, the Sarcopenia Definition and Outcomes Consortium (SDOC) recommended including weakness (low grip strength) and slowness (slow gait speed) in the definition of sarcopenia [4], and the revised consensus statement from the European Working Group on Sarcopenia in Older People defines low muscle strength as an independent indicator of probable sarcopenia [5].

Several reviews highlight the importance of protein in the prevention and management of sarcopenia [6,7,8] and frailty [9], or in the general maintenance of muscle health in old age [10,11]. A Danish longitudinal study reported that protein and leucine intake were associated with better maintenance of lean body mass after a six-year follow-up in those older than 65. However, no effect was found in younger people [12]. The nutrient requirements of older adults are poorly understood, particularly in the very old (≥85 years old), which is a heterogeneous group with widespread multimorbidity and polypharmacy [13]. Adults over 65 years consume less protein than younger ones do [12], and protein intake decreases with age [14]. Malnutrition will weaken the elderly and lead to a further decline in physical function, representing a vicious circle and decreased ability to maintain daily life activities.

Muscle mass is determined by the balance of two cellular processes, muscle protein synthesis (MPS), and muscle protein breakdown (MPB). The loss of muscle mass in ageing seems to be associated with a reduced MPS, rather than an increased MPB [15]. MPS requires a pool of available amino acids for the de novo protein synthesis. In addition to providing amino acids, protein intake initiates protein translation and stimulates the onset of MPS via the mammalian target of Rapamycin (mTOR) pathway. The amino acid leucine is especially important for the stimulation of MPS [16,17]. However, there seems to be a protein threshold, that is, a certain level of protein intake needed to initiate MPS, and this threshold seems to be higher in older people. Wall and colleagues compared post-absorptive and post-prandial muscle protein synthesis in young and older men, indicating that protein synthesis after the ingestion of 20 g of protein was 16% lower in the elderly [18]. This indicates that older people need more protein in total, and that the protein intake should preferably be evenly distributed over several meals for optimal stimulation of muscle protein synthesis multiple times during the day [16,19]. The current recommended daily allowance (RDA) of protein intake can be too low to meet the optimal protein dose to stimulate MPS, especially in the elderly [7].

Nutrients associated with prevention and management of sarcopenia are, in addition to proteins and amino acids, vitamin D, antioxidants, and long-chain polyunsaturated fatty acids (LC-PUFA) [11]. Seafood, a source of all of these nutrients, is also a source of high-quality proteins and is especially rich in leucine [20]. Thus, fish and other seafoods are a matter of interest for muscle health in ageing. Fish protein may have other properties than other animal proteins, as shown in a crossover intervention study with iso-caloric and nutrient-balanced meals, where fish protein lowered risk factors of cardiovascular diseases, compared to other animal protein sources [21]. The Norwegian food authorities recommend eating a minimum of 2–3 dinner portions of fish per week, or approximately 300–450 g fish/week [22]. To our knowledge, no studies have related habitual seafood consumption to physical function in older people.

Compared to only a couple of decades ago, more people are now reaching advanced ages, as more effective treatments and healthcare are available. In 2035, it is estimated that more than 25% of the European population will be over 65 years old [23]. Loss of physical function and strength in the elderly may decrease the ability to maintain daily life activities and increase their need for help. Thus, exploring effective strategies to counteract sarcopenia and frailty development is crucial.

The aims of this study were to assess protein intake, seafood consumption and protein mealtime distribution in elderly people, and to explore possible associations with physical performance, handgrip strength and gait speed.

## 2. Materials and Methods

### 2.1. Study Population and Study Design

In this study, we used data from the baseline assessment of a randomized intervention study of marine protein hydrolysates, described in a published protocol article [24]. The participants were elderly people (65+), recruited from several municipalities on the north-west coast of Norway. People with diabetes, progressive muscle diseases, kidney failure or active cancer were excluded from participating in the study. A total of 92 participants were recruited. Data were collected in each participant’s home, using standardized interviewer-administered questionnaires, a standardized 24-food recall method, and physical test batteries. The same researcher performed all of the data collection.

### 2.2. Data Collection

#### 2.2.1. Protein Intake and Mealtime Distribution

A 24-h multiple pass recall (MPR) was conducted, following the five-step protocol described by Moshfegh et al. [25], recording all food items and amounts eaten the day before the visit. To adapt the method to Norwegian participants, we used the food-portion illustrated booklet with a corresponding list of weights from the Norwegian study *Ungkost–2000* [26]. Energy and protein intake per day and per meal (breakfast, lunch, dinner and the evening meal) were calculated, using the diet tool from the Norwegian Directorate of Health and the Norwegian Food Safety Authority (www.kostholdsplanleggeren.no). The food database in the Diet Planner is based on the Norwegian Food Composition Table, which provides an overview of the content of energy and nutrients for the most common foods eaten in Norway.

Protein intake was expressed in both grams and grams per kilogram (kg) of body weight (BW), using adjusted BW for individuals with a body mass index (BMI) indicating under- or overweight (<22.0 kg/m^2^ or >27 kg/m^2^ in older adults [27]). In these individuals, protein intake was related to the BW corresponding to a BMI of 22 or 27 kg/m^2^, respectively. This adjustment assessed intake related to protein requirement rather than BW in under- or overweight individuals, as underweight persons require extra protein to build muscle tissue, while the ‘extra weight’ in overweight persons is often adipose tissue [28,29].

Protein mealtime distribution was assessed for each participant as the coefficient of variation (CV) of protein intake from breakfast, lunch, dinner and the evening meal (CV meals). CV meals is defined as the standard deviation (SD) of grams of protein per meal divided by grams of total protein that day. Lower values of CV meals reflect a more even distribution of protein intake across meals [30].

#### 2.2.2. Seafood Consumption

As seafood might be eaten infrequently, we used questions from a previously validated questionnaire [31] to assess habitual intake of seafood over the previous six months. The frequencies of intake of fish or other seafood for dinner and in the form of a spread, in salads, as a snack, or similar were recorded. Frequency responses were reported as never, <1 time/month, 1–3 times/month, once/week, 2–3 times/week and ≥4 times/week. The midpoint of categories was used to calculate a frequency index of meals per week. For example, 2–3 times per week was regarded as 2.5. Frequency of seafood as a spread, in salads, as a snack or similar was divided by six, as six portions of seafood as a spread correspond to one dinner portion. Thus, the summarized seafood index corresponds to dinner portion equivalents per week. This method of making a continuous scale of seafood consumption was developed and validated against biomarkers by Markhus et al. [32].

#### 2.2.3. Short Physical Performance Battery (SPPB) and Gait Speed

Physical performance was measured with the Short Physical Performance Battery (SPPB), a test developed originally for the Established Populations for Epidemiologic Studies of the Elderly, and translated into Norwegian in 2013 by Bergh and colleagues [33]. The test battery includes standing balance, walking speed and repeated chair rise. Each of the three domain scores range from 0–4 points, yielding an integer score ranging from 0–12 points. A higher score indicates a higher level of functioning, and a change of one unit is considered a clinically meaningful change [34]. A systematic review of instruments assessing performance-based physical function in older community-dwelling persons concluded that the SPPB is highly recommended in terms of validity, reliability and responsiveness [35]. The SPPB complements self-reported disability and may predict mortality and nursing home admission even at the high end of the functional spectrum [36]. Gait speed was assessed as meters per second from the 4 m walking test included in SPPB. Gait speed alone is reported to be nearly as useful as the full battery for predicting disability [37].

#### 2.2.4. Handgrip Strength

Handgrip strength was measured to the nearest 0.1 kg using the Jamar Plus+ digital hand dynamometer (Patterson Medical/Sammons Preston, Brookfield, CT, USA). The maximum out of six measured values was used for analysis [38]. Grip strength is a useful and simple measure of muscle strength. It correlates with leg strength, is a clinical marker of poor mobility, and is considered a better predictor of clinical outcomes than low muscle mass [1]. Dodds et al. provide normative data with centile values for grip strength across the life course [38].

#### 2.2.5. Other Variables

BW was recorded in light clothing to the nearest 0.1kg using a digital scale (Seca 803, Seca Hamburg, Germany) and with 1 kg subtracted to estimate subject weight without clothes. Height was measured to the nearest 0.1cm with a tape measure (Seca 201) and BMI was calculated as kg/m^2^.

The following variables, regarded as explanatory factors, were collected in the interviewer-administered questionnaires: age, gender, education, frequency of strength training, and living alone or with a partner. Age was recorded as years, and further dichotomized to 65–84 or ≥85 years to assess nutritional differences between the elderly and the very old participants. Education was also dichotomized to those with a university degree (higher education) or not. Frequency of strength training was recorded in predefined categories and dichotomized to participants who reported strength training at least once a week, and those who never or seldom did strength training.

### 2.3. Statistical Analysis

Participants’ characteristics were described as means and SD or frequencies and percentages, as appropriate. Gender and age (dichotomized) differences were assessed by independent samples t-test. The associations between nutrition and physical measures were assessed by estimating bivariate and multiple linear regression models with predefined covariates: age (continuous), gender, education, living alone, strength training and BMI, in addition to the nutrition measures of protein intake, protein distribution and seafood index. Interactions between gender and mealtime protein distribution, seafood consumption, protein intake and strength training were assessed in multiple models by using Akaike’s Information Criterion (AIC), where the smaller value means a better model. All regression models were estimated for participants with no missing values in the covariates. The results with *p*-values below 0.05 were considered statistically significant. All statistical analyses were performed using SPSS v 25.

### 2.4. Ethical Approval and Registration

All subjects gave their informed oral and written consent for inclusion before they participated in the study, which was conducted in accordance with the Declaration of Helsinki and approved by the Regional Committee on Ethics in Medical Research (REK) in Mid-Norway in September 2016 with the registration ID 2016/1152.

## 3. Results

A total of 92 participants, 61 women and 31 men, answered the questionnaires, completed the 24-h food recall, and performed the physical tests. The participants were between 65 and 93 years old, with a mean age of 73.3 years. The characteristics of participants are shown in Table 1.

Descriptive statistics for intake of energy and protein, mealtime distribution of protein, and seafood consumption for all participants, stratified by age (dichotomized) and gender, are shown in Table 2. There were no significant gender differences in any of the nutritional measures, except in protein intake (*p* = 0.01). The intake of seafood was lower in participants who were 85 years or older (*p* = 0.03). However, there were no other significant differences in nutritional measures between the two age groups. The percentage of men was similar in both age groups, namely 33.8% in the age group 65 to 84 years and 33.3% in the ≥85 years group. The seafood index in this study differed from 0.6–7.6 representing a seafood consumption frequency from every other week to daily. The mean seafood index was 3.3, that is, more than 3 times a week. None of the participants reported never eating seafood. CV meals were 0.6 ± 0.3 and differed from 0 to 1.16.

Intake of protein per meal is illustrated in Figure 1. All participants ate breakfast; however, three participants skipped dinner, while 27 and 24 participants skipped lunch and the evening meal, respectively. The protein intake was highest in the dinner meal, with a mean protein intake of 34.6 g. Protein intake per meal related to BW was 0.25 ± 0.13, 0.21 ± 0.13, 0.50 ± 0.24 and 0.18 ± 0.12 g/kg for breakfast, lunch, dinner and evening meal, respectively.

Physical performance, grip strength and gait speed of the participants are shown in Table 3. Women had lower grip strength compared to men (mean difference −18.2 kg, 95% CI −21.6; −14.8, *p* < 0.001), but not significantly different SPPB (mean difference −0.6, 95% CI −1.8; 0.5) or gait speed (mean difference −0.03, 95% CI −0.2; 0.1). The distribution of the SPPB test values was skewed, with 77.2% of the participants reaching a high score (10–12 points), and 48.9% reaching the top score of 12 points.

Associations between nutrition and physical measures are shown in Table 4. Two participants had missing MPR data. Additionally, two participants had one or more values missing on covariates and were excluded from regression analysis, leaving a total of *n* = 88.

In the bivariate models, lower age, higher education, living with a partner, and more strength training were associated with higher SPPB. None of the interactions were left in the multiple model according to AIC. In the multiple model, the associations with living with a partner and strength training were not significant. However, men had significantly higher SPPB than women. The nutrition factors were not significantly associated with SPPB. The multiple model for SPPB explained 68.9% of the total variance.

In the bivariate models for grip strength, lower age, being a man as compared to a woman and living with a partner were associated to higher grip strength. However, only age and gender remained significant in the multiple model. According to AIC, no interactions were retained in the multiple model. Nutrition factors were not significantly associated with grip strength, and the multiple model explained 76.0% of the variance.

Lower age, living with a partner, and more strength training were significantly associated with higher gait speed in the bivariate models. In the multiple model, lower age, being a man as compared to a woman, and more strength training were significantly associated to higher gait speed. The model contained an interaction between gender and seafood consumption. This interaction was not significant, but not ignorable according to AIC. Exploring the interaction further showed that higher seafood consumption was associated with higher gait speed among women (*p* = 0.03), while this association was not significant among men. However, the overall gender differences regarding this association were not significant (*p* = 0.07 for interaction). The multiple model explained 60.0% of the variance.

## 4. Discussion

The participants in this project had a mean protein intake of 71.3 g and 83.7 g per day for women and men, respectively. Protein intake related to BW were 1.1 g/kg for both genders. Recommended Dietary Allowance (RDA) of protein is set to 0.8 g protein per kg of BW by the Institute of Medicine (IOM). However, to prevent the loss of muscle mass and function in ageing, a higher intake of at least 1.0–1.2 g/kg/day was recommended by the Society on Sarcopenia, Cachexia and Wasting Disorders (SCWD) [40], the European Society for Clinical Nutrition and Metabolism (ESPEN) [41], and the PROT-AGE Study Group [42]. Current national recommendations for protein intake vary from 0.8 g/kg/day (WHO, UK, US guidelines) to as much as 1.3 g/kg/day (Nordic Nutrition Recommendations) [10]. Thus, the protein intake in our study was in line with current recommendations. However, other Norwegian studies have reported higher protein intakes, for example, Norkost3 [43] and the Tromsø study 2015–2016 [14]. The protein intake in our study was more in line with intakes reported in studies with older participants from other countries [30,44,45] and with the oldest age group (80+) in the Tromsø study [14].

The participants in our study reported a mean energy intake of 1734 kcal and 1943 kcal in women and men, respectively. A reduction in physical activity, and old age itself, may reduce the need for energy, that is, the anorexia of ageing. Although both the energy need and intake decrease, the need for nutrients remains stable or even increases with age, and the portion of protein should be increased accordingly to a decreased energy intake in subjects with an energy intake below 8 MJ (1911 kcal) [46]. Thus, older people need nutrient-dense food to meet the nutrient recommendations. In this way, the concept of “healthy” food changes over the lifespan, as nutritional needs and energy expenditure change. The elderly might be highly conscious of a healthy lifestyle but still following a diet recommended to them while they were younger, leading to malnutrition in old age.

Despite the well-documented satiating effect of protein, improved protein intake did not result in lower total energy intake in a trial in a rehabilitation centre, where participants were allocated either normal or protein-enriched bread and yoghurt [47]. Therefore, increasing protein content in breakfast or lunch is a feasible way to increase protein intake and counteract the blunted MPS response seen in ageing. Murphy et al. recommend a per meal protein intake of 0.4 g/kgBW [39]. According to this recommendation, the protein intake in our study (1.1 g/kg/day) could yield a maximum of two meals exceeding 0.4 g/kg. The distribution of protein intake in our sample was skewed, meaning most of the protein was consumed at the dinner meal, and dinner was the only meal exceeding a mean of 0.4 g/kg. Other studies have also found a skewed protein intake, for example, among older adults in the UK and US where the evening dinner meal contains 40–50% of the total protein intake [39]. However, compared to the Norwegian meal pattern, other countries often have a higher protein consumption at lunch than the participants in our study [28,39]. The Norwegian lunch is often quite similar to the breakfast meal with bread with some kind of spread, and it is not common to consume a warm meal for lunch. Having a warm meal for lunch is associated with a higher protein intake, as bread-based meals tend to be less protein-rich [48].

To our knowledge, no previous studies have described per meal protein distribution in Norwegian populations. The protein mealtime distribution (CV meals) in this study was 0.6 ± 0.3 for both women and men. Bollwein et al. found a more even protein eating pattern in non-frail participants than in frail and pre-frail participants [30]. However, protein intake was less distributed during the day (higher CV meals) in all groups (frail, pre-frail and non-frail) compared to our study. Even the non-frail participants in the Bollwein study had a CV meal of 0.68 [30]. The individual CV meals in our study varied from 0 (all meals with an equal amount of protein) to 1.16. A good distribution of protein across meals does not necessarily have an effect if each meal is insufficient for stimulating MPS [49]. A good protein distribution could, on the contrary, have a negative effect, as it might cause none of the individual meals to reach the required protein intake to initiate MPS. Previous research has shown that the protein dose needed to stimulate MPS is about 20 g protein per meal, or 35–40 g for older people, as older muscle has an attenuated ability to transduce signals to MPS [49]. Murphy et al. [47], Trayler et al. [7] and Franzke et al. [10] argue for a per meal recommendation of protein to stimulate MPS in older adults. However, this population is heterogeneous and several aspects with both the individuals (e.g., digestion and anabolic response) and with the diet (e.g., protein quality and leucine content) influence the MPS response to a meal. Thus, further research is necessary before the concept of per meal recommendation is expedient and practically useful.

We found no association between CV meals or protein intake, and physical performance or strength. The small sample size and the high number of participants with an SPPB top score weakened the possibility of detecting such associations. The mean score on the SPPB was 10.3 ± 2.7; however, 48.9% of the participants reached the top score of 12 points. The ceiling effect in SPPB, also described by Bergland et al. [50], reduced the possibility of detecting associations between nutrition and physical function. Mean grip strength was 44.4 ± 9.4 and 26.2 ± 6.8kg in men and women, respectively. Compared to the normative data provided by Dodds et al. [38], the participants in this study are stronger than average, and their mean grip strength is more in line with a population of 60 years old. Mean gait speed was 1.0 ± 0.3 m/s, which is high compared to other studies [37]. No significant associations were found between the nutrition factors and physical performance or strength; however, this might be caused by a ceiling effect in the chosen test batteries.

The consumption of fish or other seafood in this study was high, and most subjects had an intake in accordance with the recommendation of 2–3 meals per week. As the participants were recruited to a trial of fish protein hydrolysate, one could assume that our sample was especially interested in seafood and its health advantages. However, a comparable high fish consumption was described in the large community-based Hordaland Health Study (HUSK) [51,52]. In the HUSK study, fish intake was reported as grams per 1000 kcal, and the mean intake was 45 and 51 g/1000kcal in women and men, respectively. This intake corresponds to approximately 500–700 g/week, or 3–4 portions per week [53]. These studies are comparable to our study, as both include elderly people living on the coast of Norway [51,52]. Fish consumption is traditionally high in these areas.

Seafood consumption was significantly lower in the very old, that is, participants who were ≥85 years. As the sample of very old participants was low (*n* = 16), this is a weak finding. However, more research should focus on diet in the very old. Our experience from data collection was that the oldest participants more often ate precooked meals, and this may explain the low seafood consumption. More use of precooked meals was also reported elsewhere and may affect the nutritional status of older people [54].

A higher seafood consumption was associated with higher gait speed in women but not in men, and not in the overall sample. No causal effects can be derived as this is a cross-sectional study; however, this finding raises interesting questions. Women who have higher seafood intake might also be more health conscious in other ways, for example, engaged in more exercise. However, association between fish consumption and healthy lifestyle did not differ between men and women in a large Swedish study [55]. Another interesting question is whether women and men derive different health benefits from seafood. Gender differences have previously been described in per meal protein need, with women requiring more protein than men do, so as to trigger postprandial MPS [49]. Bearing in mind the importance of leucine in the stimulation of MPS, a diet high in seafood consumption could be beneficial in optimizing protein quality and thus enhance MPS in women.

## 5. Conclusions

This study did not show any associations between protein or seafood consumption, or protein distribution and physical performance or strength. The participants in the study were generally of good health, and possible associations with physical performance might not have been detected due to the ceiling effect of our chosen test batteries. However, the results of this study show the need for more research focused on gender differences in the relationship between nutrition and physical performance. More research is needed reflecting seafood consumption in the very old and to determine if a good mealtime distribution of protein is beneficial for muscle health, or if a per meal recommendation of protein intake is more useful.

## Figures and Tables

**Figure 1 geriatrics-05-00100-f001:**
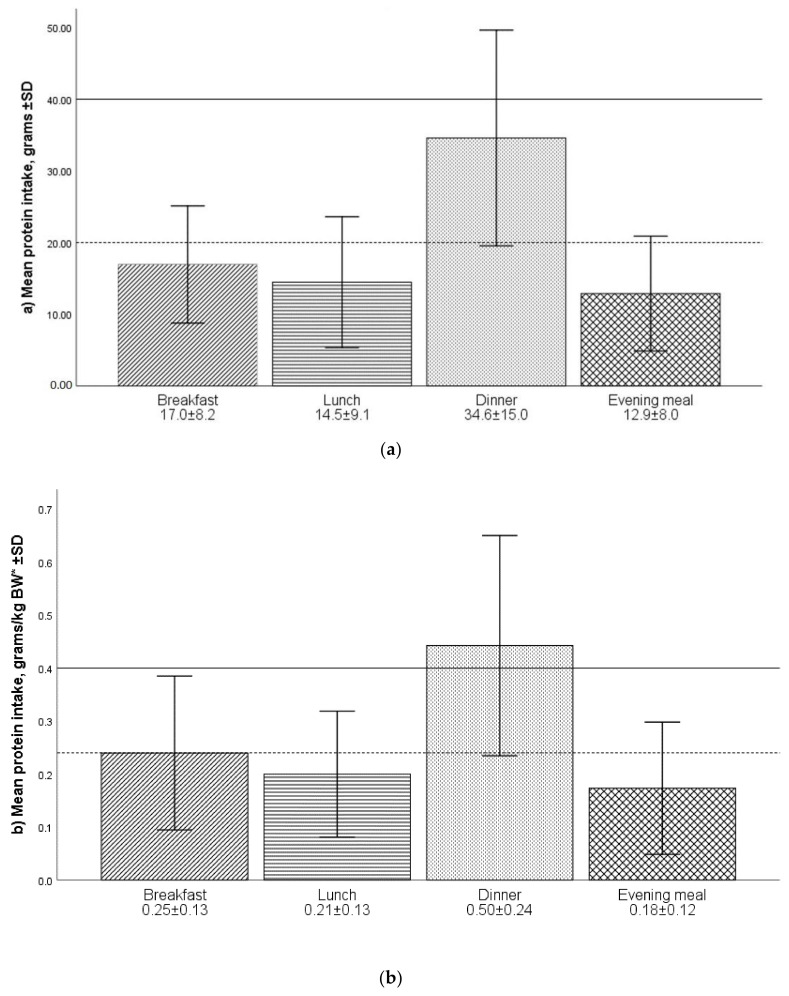
(**a**,**b**) Protein intake (mean ± SD) estimated from 24-h recall for breakfast (*n* = 90), lunch (*n* = 63), dinner (*n* = 87) and evening meal (*n* = 66), shown as (a) grams and (b) grams per kg bodyweight (* bodyweight adjusted in underweight (BMI < 22) or overweight (BMI > 27) individuals). Vertical reference lines represent intake shown to initiate muscle protein synthesis (MPS) in young (dotted line) and old (full line) adults as described in Murphy et al. [39]. Error bars represent ± 1SD. The number of participants varies between meals as not all participants ate four meals per day.

**Table 1 geriatrics-05-00100-t001:** Baseline characteristics of study participants.

	*n*		Min–Max
Age, mean years ± SD	92	73.8 ± 8.9	65–93
Male gender, *n* (%)	92	31 (33.7)	
Have higher education, *n* (%)	92	41 (44.6)	
Living alone, *n* (%)	92	34 (37.0)	
Strength training ≥1/week, *n* (%)	92	43 (46.7)	
Body mass index (BMI), mean kg/m^2^ ± SD	91	25.9 ± 4.5	16.6–39.9

**Table 2 geriatrics-05-00100-t002:** Energy and protein intake and seafood consumption for all participants, stratified by gender and age, *n* = 90.

		Energy Intake (kcal)	Protein Intake (g)	Protein Intake (g/kg) ^1^	CV Meals ^2^	Seafood Index ^3^
All	All, mean ± SD	1804 ± 529	75.4 ± 22.6	1.1 ± 0.4	0.6 ± 0.3	3.3 ± 1.3
Gender difference	Women (*n* = 60), mean ± SD	1734 ± 535	71.3 ± 23.4	1.1 ± 0.4	0.6 ± 0.3	3.3 ± 1.4
	Men (*n* = 30), mean ± SD	1943 ± 496	83.7 ± 18.7	1.1 ± 0.2	0.6 ± 0.3	3.2 ± 1.1
	Mean difference	−208.9	−12.4	0.04	0.05	0.08
	95% CI	−441.1; 23.4	**−22.2; −2.6**	−0.1; 0.2	−0.1; 0.2	−0.5; 0.7
	*p*-value	0.08	**0.01**	0.57	0.40	0.78
Age difference	Age 65–84 (*n* = 74), mean ±SD	1814 ± 563	76.8 ± 24.0	1.1 ± 0.4	0.6 ± 0.3	3.4 ± 1.3
	Age ≥85 (*n* = 16), mean ± SD	1759 ± 342	69.0 ± 13.7	1.1 ± 0.3	0.6 ± 0.3	2.7 ± 1.0
	Mean difference	54.4	7.8	−0.05	−0.008	0.7
	95% CI	−236.9; 345.7	4.4; −1.2	−0.2; 0.2	−0.2; 0.2	**0.1; 1.4**
	*p*-value	0.71	0.09	0.65	0.92	**0.03**

^1^ Gram per kg bodyweight, with bodyweight adjustment in underweight (BMI < 22) or overweight (BMI > 27) individuals. ^2^ Coefficient of variance in protein mealtime distribution. ^3^ Index representing dinner portion equivalents per week.

**Table 3 geriatrics-05-00100-t003:** Physical performance (Short Physical Performance Battery (SPPB) and gait speed) and grip strength in all participants, and men and women, respectively, *n* = 92.

		Mean ± SD	Min–Max
SPPB ^1^, score	All participants	10.3 ± 2.7	2–12
	Men	10.7 ± 2.1	5–12
	Women	10.0 ± 2.9	2–12
Grip strength, kg	All participants	32.3 ± 11.6	7.8–57.6
	Men	44.4 ± 9.4 *	20.9–57.6
	Women	26.2 ± 6.8 *	7.8–40.3
Gait speed, m/s	All participants	1.0 ± 0.3	0.2–1.9
	Men	1.0 ± 0.3	0.4–1.9
	Women	1.0 ± 0.3	0.2–1.6

^1^ Short Physical Performance Battery. * Significant gender difference (*p* < 0.001).

**Table 4 geriatrics-05-00100-t004:** Results of linear regression assessing the association between the explanatory factors protein mealtime distribution (CV meals), seafood consumption and protein intake, and the outcome measures short physical performance battery (SPPB), grip strength and gait speed, controlled for the predefined covariates age, gender, education, living alone or with a partner, strength training and body mass index (BMI), *n* = 88.

	SPPB ^1^				Grip Strength				Gait Speed			
	Bivariate models		Multiple Model		Bivariate Models		Multiple Model		Bivariate Models		Multiple Model	
	Regr. Coeff. (95% CI)	*p*-value	Regr. Coeff. (95% CI)	*p*-value	Regr. Coeff. (95% CI)	*p*-value	Regr. Coeff. (95% CI)	*p*-value	Regr. Coeff. (95% CI)	*p*-value	Regr. Coeff. (95% CI)	*p*-value
Age	−0.2 (−0.3; −0.2)	<0.01	−0.2 (−0.3; −0.2)	<0.01	−0.5 (−0.8; −0.3)	<0.01	−0.5 (−0.7; −0.4)	<0.01	−0.02 (−0.03; −0.02)	<0.01	−0.02 (−0.03; −0.02)	<0.01
Male gender	0.7 (−0.5; 1.9)	0.27	1.1 (0.4; 1.9)	0.01	18.2 (14.6; 21.7)	<0.01	18.9 (15.9; 21.9)	<0.01	0.02 (−0.1; 0.2)	0.73	0.3 (0.1)	0.02
Education, higher	1.6 (0.5; 2.7)	0.01	0.9 (0.2; 1.6)	0.02	0.7 (−4.3; 5.6)	0.78	0.4 (−2.4; 3.1)	0.8	0.1 (−0.002; 0.3)	0.05	0.04 (−0.06; 0.1)	0.42
Living alone	−1.7 (−2.8; −0.6)	<0.01	0.04 (−0.8; 0.9)	0.93	−9.9 (−14.5; −5.2)	<0.01	0.1 (−3.2; 3.3)	0.97	−0.2 (−0.3; −0.03)	0.02	0.02 (−0.09; 0.1)	0.77
Strength training ^2^	1.2 (0.1; 2.3)	0.04	0.6 (−0.2; 1.4)	0.11	−2.4 (−7.3; 2.5)	0.34	0.1 (−2.8; 3.0)	0.96	0.2 (0.1; 0.3)	< 0.01	0.1 (0.02; 0.2)	0.02
BMI	0.1 (−0.1; 0.2)	0.4	0.1 (−0.04; 0.2)	0.22	0.3 (−0.3; 0.8)	0.29	0.2 (−0.2; 0.5)	0.32	−0.003 (−0.02; 0.01)	0.63	−0.005 (−0.02; 0.01)	0.44
CV meals ^3^	0.7 (−1.4; 2.7)	0.52	1.0 (−0.3; 2.2)	0.14	−2.3 (−11.0; 6.5)	0.61	2.2 (−2.7; 7.0)	0.38	0.1 (−0.1; 0.3)	0.34	0.08 (−0.09; 0.2)	0.34
Seafood index ^4^	0.4 (−0.1; 0.8)	0.1	0.1 (−0.2; 0.4)	0.38	1.3 (−0.6; 3.2)	0.19	1.0 (−0.1; 2.0)	0.07	0.05 (−0.003; 0.09)	0.07	0.1 (0.06)	0.03
Protein intake ^5^	0.3 (−1.3; 1.2)	0.73	0.5 (−0.6; 0.4)	0.4	−1.0 (−7.9; 5.9)	0.78	0.8 (−3.3; 4.9)	0.7	−0.04 (−0.2; 0.1)	0.67	−0.09 (−0.2; 0.05)	0.2
	Gender × seafood index interaction										−0.08 (0.04)	0.07

^1^ Short Physical Performance Battery. ^2^ Perform strength training ≥ once a week. ^3^ Coefficient of variance in protein mealtime distribution. ^4^ Index representing dinner portion equivalents per week. ^5^ Protein intake g/kg Bodyweight, adjusted in underweight (BMI < 22) or overweight (BMI > 27) individuals.

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
