# Peer review of "Protein Intake, Protein Mealtime Distribution and Seafood Consumption in Elderly Norwegians: Associations with Physical Function and Strength"

_geriatrics, 2020, doi:10.3390/geriatrics5040100_

Round 1

Reviewer 1 Report

Dear Authors and Editor, It is an interesting study and congrats the authors for it, I will leave my comments and suggestions below:

  • Please, I suggest a professional English review, mainly in the Introduction part;
  • Abstract, I suggest presenting the study highlights, some outcomes presented in this session were just participants descriptive data as ¨Mean score on SPPB was 10.3±2.7, and 48.9% of the participants reached the 24 top score of 12 points. Mean grip strength was 44.4±9.4 kg (men) and 26.2±6.8 kg (women). Mean 25 gait speed was 1.0±0.3 m/s.¨;
  • P2, line 71 - ¨…compared to animal protein sources…¨; I suggest writing other animal protein sources, because fish is an animal too;
  • P5, Table 2, the number 1, reference about Gram per kg bodyweight, it has to be out of the parentheses ( )1 on top of the table; Because, it can make a misunderstanding of units relation;
  • P5 and 6 – Figure 1, The written part about ¨Vertical references…muscle¨ is not clear, where is the reference citation indicating the MPS values? And, I suggest changing ¨muscle¨ to ¨adults¨, because ¨young and old muscle¨ sounds not correctly; The outcomes are expressed in mean and SD? Please, add in the legend. I suggest to add at same figure, protein intake data normalized per bodyweight, and add another axis (Gram/Kg of bodyweight) graph right side;
  • P6, Table 3 and Legend: It has to explain, that first line, on each topic is all participants mean and SD; Please, add proper symbol to express significant statistical difference (e.g. *), not number, because it was presented number 1 for SPPB and 2 for statistical difference in gender; Please, check the gait speed values, because they are the same for all three (all participants, Men, and Women); And it is quite impressive the grip strength values for Older Men in this study, mean 44.4 kg, and maximum 57.6kg, in my experience I just saw some high values in young adults strength training practitioners;
  • I felt information lack about Protein type ingestion, animal or plant protein sources, for the participants; It could be expressed by percentage; It could be interesting to discuss about essential amino acids ingestion and MPS.

Reviewer 2 Report

This study investigated possible associations between protein or seafood consumption, or protein distribution and physical performance or strength. The subjects are elderly men and women in Norway.

The quality of writing is excellent.

Keywords,  delete per meal recommendation; older. Add older adults (or a similar term). SPPB should probably be replaced by a more commonly used term.

Line 136, clarify 4 m (4 minutes?)

This study is cross-sectional. However, that term is not mentioned in the paper and nor is the possible errors that can arise from that design. In particular, it is impossible to say if any association are causal. For example, on line 317 the authors say “A higher seafood consumption was associated with higher gait speed in women but not in men… This finding raises the question whether women and men derive different health benefits from seafood.” An alternative explanation is that many women who are health conscious engage in more exercise (and therefore have higher gait speed) and also eat a healthier diet (including more seafood).

Round 2

Reviewer 1 Report

Dear Authors and Editor, my comments below:

  • The authors attended for most of my suggestions to improve the manuscript, and they responded properly;
  • I just suggest English review at Introduction part;

Reviewer 2 Report

I am now satisfied with the paper.